# Biological Specimen Banking as a Time Capsule to Explore the Temporal Dynamics of Norovirus Epidemiology

**DOI:** 10.3390/v15122303

**Published:** 2023-11-24

**Authors:** Floriana Bonura, Chiara Filizzolo, Mariangela Pizzo, Giuseppa L. Sanfilippo, Federica Cacioppo, Emilia Palazzotto, Francesca Di Bernardo, Antonina Collura, Vito Martella, Simona De Grazia, Giovanni M. Giammanco

**Affiliations:** 1Dipartimento di Scienze per la Promozione della Salute, Materno-Infantile, di Medicina Interna e Specialistica di Eccellenza “G. D’Alessandro”, Università di Palermo, Via del Vespro 133, 90127 Palermo, Italy; chiara.filizzolo@gmail.com (C.F.); mariangela.pizzo22@gmail.com (M.P.); sanfilippogiusi@gmail.com (G.L.S.); federica.cacioppo03@unipa.it (F.C.); emiliapalazzotto@gmail.com (E.P.); simona.degrazia@unipa.it (S.D.G.); giovanni.giammanco@unipa.it (G.M.G.); 2Unità Operativa di Microbiologia e Virologia, Ospedale Civico e di Cristina, ARNAS, 90129 Palermo, Italy; francesca.dibernardo@arnascivico.it (F.D.B.); antonina.collura@arnascivico.it (A.C.); 3Dipartimento di Sanità Pubblica e Zootecnia, Università Aldo Moro di Bari, 70010 Valenzano, Italy; vito.martella@uniba.it

**Keywords:** norovirus, evolution, genotypes, GII.4, GII.3, GII.2, Italy

## Abstract

Norovirus is recognised as a major cause of epidemic and sporadic acute gastroenteritis (AGE) in all age groups. Information on the genetic diversity of the noroviruses circulating in the 1980s and 1990s, before the development and adoption of dedicated molecular assays, is limited compared with the last decades. Between 1986 and 2020, uninterrupted viral surveillance was conducted in symptomatic children hospitalized with AGE in Palermo, Italy, providing a unique time capsule for exploring the epidemiological and evolutionary dynamics of enteric viruses. A total of 8433 stool samples were tested using real-time RT-PCR. All samples were stored at −20 or −80 °C until processing. In this 35-year long time span, noroviruses of genogroup II (GII) were detected in 15.6% of AGE requiring hospitalization, whilst GI noroviruses were detected in 1.4% of AGE. Overall, the predominant norovirus capsid (Cap) genotype was GII.4 (60.8%), followed by GII.3 (13.3%) and GII.2 (12.4%). Temporal replacement of the GII.4 Cap variants associated with different polymerase (Pol) types were observed over the study period. The chronology of emergence and circulation of the different GII.4 variants were consistent with data available in the literature. Also, for GII.3 and GII.2 NoVs, the circulation of different lineages/strains, differing in either the Cap or Pol genes or in both, was observed. This long-term study revealed the ability of noroviruses to continuously and rapidly modify their genomic makeup and highlights the importance of surveillance activities in vaccine design.

## 1. Introduction

Noroviruses are a major cause of acute gastroenteritis (AGE) worldwide. Noroviruses belong to the family *Caliciviridae* and are non-enveloped viruses with a positive-sense single-stranded RNA genome (7.3–7.5 kb) organized into three open reading frames (ORFs). ORF1 encodes a large non-structural polyprotein that includes the viral RNA-dependent RNA polymerase (RdRp), whilst ORF2 encodes the major capsid protein (VP1) and ORF3 encodes the minor capsid protein (VP2) [1]. Noroviruses were first identified upon electron microscopy observation of stool samples from patients involved in an outbreak of AGE at an elementary school in Norwalk, OH, USA in 1968, and the causative agent was therefore named Norwalk virus [2]. With the development of specific molecular assays in the 1990s, the role of norovirus as a causative agent of AGE has been clarified [3]. Noroviruses are responsible for more than 90% of non-bacterial AGE epidemics worldwide and are considered the first or second most important cause of diarrhoea in children, along with rotaviruses [4,5]. Noroviruses have been estimated to cause around 1.1 million hospitalizations and up to 200,000 deaths per year, mostly in children less than 5 years of age in developing countries [4,5]. The genetic diversity of noroviruses is a challenge for diagnostics, classification, and the development of vaccines. Based on the complete amino acid (aa) sequence of VP1, noroviruses are classified into ten genogroups (GI–GX) and 49 capsid (Cap) genotypes. A total of 60 confirmed polymerase (Pol) types have also been described based on partial nucleotide (nt) sequences of the polymerase region [6]. The majority of norovirus strains associated with diseases in humans belong to genogroups GI and GII and are further classified into more than 40 human genotypes [7]. Multiple norovirus genotypes co-circulate in human populations, but GII genotype 4 (GII.4) has been associated with most outbreaks (>80%) and sporadic cases of gastroenteritis in both developed and developing countries [8,9,10]. Since the mid-1990s, six global epidemics of norovirus GII.4 have been documented and each has been associated with periodic emergence of novel GII.4 variants at intervals of 3–4 years. The pandemic GII.4 variants include US95_96, which emerged in the late 1990s [11,12], followed by Farmington_Hills_2002 in 2002 [13,14], Hunter_2004 in 2004 [15], Den Haag_2006b in 2007 [16,17], NewOrleans_2009 in 2009 [18] and finally Sydney_2012 in 2011–2012 [19,20]. It has been proposed that new pandemic GII.4 NoV variants generally evolve through the acquisition of residue substitutions in the capsid protein VP1 that alter antigenicity, enabling evasion of host immunity [19,21,22,23,24,25,26] and/or by modifying affinity to histo-blood group antigen (HBGA) receptors [27]. In addition to the pandemic GII.4 variants of global relevance, minor GII.4 variants have been described in epidemics restricted to specific geographical regions, namely the variants Asia_2003 [28], Yerseke_2006a [16], Osaka_2007 [29] and Apeldoorn_2008 [30]. Norovirus genotyping has been complicated by the emergence of recombinant strains that have polymerase and capsid regions derived from separate ancestral strains [20]. The global molecular epidemiology of emerging GII.4 strains is largely based on data from outbreak surveillance programmes that were enacted worldwide in the 2000s and 2010s. Improvements in diagnostics, with the development and large adoption of molecular assays for noroviruses, have provided valuable information on norovirus epidemiology in the last two decades, but information on the diversity of noroviruses in the 1980s and 1990s is limited. Uninterrupted surveillance for AGE in hospitalized children has been carried out in Palermo, Italy, since the mid-1980s, providing a unique collection spanning more than 35 consecutive years that can be used as a time machine to investigate retrospectively the genetic evolution of enteric viruses. The present study summarises a more than three-decades-long surveillance, offering a useful temporal observatory of molecular epidemiology based on sequence data on norovirus strains circulating in the local paediatric population of Palermo since the end of the last century. 

## 2. Material and Methods

Over 35 consecutive years, from 1986 to 2020, uninterrupted norovirus surveillance was conducted in Palermo, South of Italy. A total of 8433 stool samples were collected from paediatric patients (<5 years old) hospitalized with AGE at the “G. Di Cristina” Children’s Hospital. AGE was defined by at least 3 watery stools with or without bouts of vomiting over 24 h and lasting less than 7 days, with no identifiable symptoms other than those associated with infective gastroenteritis. Stool samples were collected within 12 h of admission to the hospital to avoid inclusion of nosocomial cases and stored at −20 or −80 °C until processing. Viral RNA was extracted from stool samples collected from 1986 to 2000 using the ELITE InGenius automated extraction platform (ELITechGroup, Inc., Bothell, WA, USA). For samples collected from 2001 to 2020, viral RNA was extracted from 140 μL of a 10% stool suspension using a QIAamp Viral RNA Mini Kit (QIAGEN, Hilden, Germany), according to the manufacturer’s instructions. Undiluted (pure) and diluted (1:10) RNA samples were used to reduce the effect of the possible presence of PCR inhibitors in stool samples. Random hexamers were used for reverse transcription reaction to obtain complementary DNA (cDNA) using MMLV reverse transcriptase (Invitrogen, Carlsbad, CA, USA). A quantitative reverse transcription (RT)-PCR assay (qRT-PCR) able to differentiate between GI and GII norovirus-positive samples was used to detect norovirus RNA [31]. Norovirus-positive specimens were genotyped using a multi-target strategy, generating sequence data for diagnostic region A (spanning the ORF1 region coding for the polymerase) and region C (encompassing the initial part of ORF2 and coding for the capsid), using primers JV12/JV13 and COG2F/G2SKR, respectively [31,32,33,34,35]. The hypervariable capsid P2 domain was tested in a selection of 40 samples representative of different GII.4 norovirus variants observed during the study period using primer EVP2F and EVP2R, as previously described [36].

Sequence alignment was performed using CLUSTAL W [37]. Phylogenetic analysis was carried out using the MEGA X software [38], with the Kimura 2-parameter model as the substitution method. Phylogenetic trees of partial sequences of Pol and Cap were constructed using the Maximum-likelihood method, with 1000 bootstrap replicates. Genotype assignment was performed using the Noronet automated genotyping tool (https://www.rivm.nl/en/noronet/databases; accessed on 1 April 2023) and the CDC calicivirus typing tool (https://calicivirustypingtool.cdc.gov; accessed on 1 April 2023).

## 3. Results

### 3.1. Prevalence and Typing of Noroviruses

Out of 8433 stool samples collected from symptomatic children hospitalized with AGE, GII norovirus was detected in 15.6% (1317/8433) of the patients, whilst GI norovirus was detected in 1.4% (117/8433) of the patients. The temporal distribution of norovirus GI and GII infections is shown in Figure 1. Seasonality could be calculated starting from 2002 since the sampling date was not available for older samples. However, sampling was less abundant and homogeneous until 2011, making estimates of seasonality for these years less reliable compared with the latest decade. Figure 2 shows the seasonality of norovirus circulation in Palermo from 2002 to 2020.

Genotyping of the Cap and Pol regions of norovirus GII was performed in 64.3% (847/1317) and 61.6% (812/1317) of the norovirus GII-positive samples, respectively. The most prevalent Cap genotype was GII.4 (60.8%), followed by GII.3 (13.3%), GII.2 (12.4%), GII.6 (4.7%), GII.17 (2%) and GII.1 (1.9%). Overall, 49.5% (652/1317) of the norovirus GII-positive samples were fully typed, providing the Cap/Pol combination. GII.4[P4] accounted for 28.3% of the fully typed strains, followed by GII.4[P16] (19.9%), GII.4[P31] (15.3%), GII.2[P16] (8.2%), GII.2[P2] (5.3%), GII.3[P21] (3.5%), GII.6[P7] (3%), GII.3[P12] (2.9%), GII.17[P17] (2.4%), GII.3[P3] (1.8%), GII.13[P16] (1.7%) and GII.3[P30](Pc) (1.4%). The other cap/pol combinations were detected in <1% of the samples. The temporal distribution of Cap and Pol genotypes and fully typed Cap/Pol strains are shown in Figure 3a, Figure 3b and Figure 3c, respectively. 

### 3.2. Focus on the Early Stages of Norovirus Circulation

Norovirus RNA was not detected in the 191 available stool samples collected from 1986 to 1988. Noroviruses were first detected in Palermo in 1989 stool samples and circulated at low frequency (up to 5.4%) until 1994, with a high genotype diversity in ORF2 (Figure 1). In particular, GII.4 and GII.6 Cap genotypes were detected in 1989 but were replaced by GII.2 in 1990 and GII.6 in 1993. In 1994, the most prevalent genotype was GII.3 (47.4%), followed by GII.2 and GII.8 (15.8%), GII.5 (10.5%) and GII.4 and GII.13 (5.3%). The GII.4 genotype was predominant from 1995, with the exception of the years 2003, 2004 and 2016 (Figure 3a). Based on sequence analysis of region A (ORF1), the GII.P4 Pol type was predominant from 1989 to 2011, with the exception of 1994, when GII.P3 (64.3%), GII.P5 (21.4%) and GII.P8 (14.3%) co-circulated (Figure 3b). 

### 3.3. Genetic Evolution and Diversification of Norovirus Genotypes 

Phylogenetic analyses based on sequences generated from the diagnostic region C (ORF2) were performed in order to decipher the genetic diversification of the three predominant norovirus Cap types, i.e., GII.4, GII.3 and GII.2, over time. 

#### 3.3.1. Analyses of GII.3 and GII.2 Noroviruses

Phylogenetic analysis showed that the Cap gene sequences of the Italian GII.3 noroviruses segregated into three different clusters (II–IV) that were previously defined by Boon et al. [39]. The GII.3 strains circulating from 1994 to 1997 segregated within Cap cluster II together with GII.3[P3] strains, which emerged in the 1980s and 1990s in Japan, USA and Mexico, and were all characterized by a P3 Pol gene. The GII.3 strains detected from 2003 onward segregated within Cap clusters III and IV and showed different Cap/Pol combinations. In particular, all the Italian GII.3 strains circulating from 2004 to 2006 and from 2014 to 2016 contained a GII.P21 Pol gene and segregated in cluster III, whilst the GII.3 noroviruses detected in 2012 and 2019 segregated within cluster IV in association with different Pol types, as follows: in 2012 they were associated with P4_2006b, P12, P16 and P21 Pol types, whilst in 2019 they were associated with P12, P16 and P30 Pol types (Figure 4b). 

The Italian GII.2 strains detected over the whole study period segregated into five different Cap lineages (a–e) in the GII.2 phylogenetic tree. GII.2 lineages generally included strains isolated in consecutive years and showing the same Pol type, except for lineage a and c where two different Pol types were included, with P2 being replaced by P16 and P34, respectively (Figure 4c).

#### 3.3.2. Analysis of the GII.4 Norovirus Variants

The first GII.4 strain was detected in Palermo in a faecal sample collected in 1989 and was characterised as a Lordsdale variant, displaying 99.8% nt identity to the reference strain (X86557) detected in the UK four years later, in 1993. Significant heterogeneity (97.4–100% nt identity) was observed among the 39 GII.4 noroviruses detected in Palermo from 1994 to 1997, segregating into different branches of the GII.4 US95_96 variant sub-tree together with contemporary noroviruses circulating in Europe and the USA in the same period (Figure 4a). In 2002, the GII.4 variant Farmington_Hills_2002 became predominant, whilst the GII.4 variant Hunter_2004 was predominant from 2004 to 2006. In 2006, the variant Den Haag_2006b emerged and co-circulated with the variant Yerseke_2006a in 2007 and with the GII.4 variant Apeldoorn_2007 in 2008. From 2009, the GII.4 variant NewOrleans_2009 became predominant for a couple of years, before being replaced by the GII.4 variant Sydney_2012, which emerged in 2011 in Palermo [40]. The GII.4 variant Sydney_2012 circulated stably until 2020 (Figure 4a). However, while the Cap Sydney_2012 was initially associated with a P31 (Pe) Pol-type, recombinant strains emerged with the NewOrleans_2009 Pol gene from 2013 onwards, followed, from 2017 onwards, by recombinant strains with a GII.P16 Pol gene (Figure 4a). 

### 3.4. Antigenic Variation in the Hypervariable P2 Domain of GII.4 Variants

Analysis of the aa sequences of the hypervariable P2 domain of the reference strains of GII.4 norovirus variants showed high aa identity between the ancestral GII.4 norovirus strain Camberwell and the older GII.4 variants (Lordsdale and US95_96) (94.9–98.1% id), whilst several aa substitutions have accumulated since 2002, with the later emergence of GII.4 norovirus variants (Figure 5). In particular, an insertion at position 394, located in the D hypervariable domain, appeared in 2002 in the Farmington_Hills_2002 variant together with several conserved aa substitutions (D298N/E, L333M/V, Q376E/D and N407S/D) in the hypervariable domains. Additional aa mutations were observed in the Den Haag_2006b and Apeldoorn GII.4 norovirus variants, emerging in 2006 and 2007 in the A epitope of the P2 domain (i.e., T368A, N372D). When the aa sequences of the Italian GII.4 strains collected over the study period were compared with the reference sequences, the Italian Lordsdale strain differed from the prototype X86557 at two residues located in the A (N372D) and E (D393E) hypervariable domains, whilst the Italian GII.4 US95_96 strain differed from the prototype AJ004864 at three residues (S309N, E340G and N393S). The sequences of the Italian GII.4 Farmington_Hills_2002, Yerseke_2006a, Den Haag_2006b and NewOrleans_2009 strains were conserved with respect to the prototype sequences (AY502023, EF126963, EF126965 and GU445325, respectively), whilst several polymorphisms were observed in the P2 epitopes of the Italian GII.P31-GII.4 Sydney_2012 and GII.P4New_Orleans/GII.4 Sydney_2012 strains, which circulated over a longer time frame (Figure 5).

## 4. Discussion

Noroviruses were first identified—using electron microscopy—as the cause of AGE in symptomatic children in 1970 [2] but remained underestimated until the development and adoption of molecular assays specific for routine diagnostics in the 1990s. In parallel, the literature on noroviruses has increased significantly since the year 2000, with an average of 30 manuscripts per year versus less than 4 manuscripts per year in the second half of the 1990s (https://pubmed.ncbi.nlm.nih.gov/?term=norovirus, searched on 1 January 2023). However, information on the epidemiology of noroviruses circulating before the 2000s is limited and fragmented [39,41,42]. 

In this archival retrospective study, the molecular epidemiology of noroviruses was investigated over 35 consecutive years, from 1986 to 2020. This archive of stool specimens and/or genetic material extracted from faecal samples derives from one of the longest enteric virus surveillances conducted in the European continent, providing an essential tool for investigating the evolution of noroviruses. 

Over the study period, norovirus infection was detected in 17% of paediatric patients (<5 years old) hospitalized in Palermo, Italy. GII noroviruses, first detected in Palermo in 1989, represented the most prevalent genogroup, accounting for 15.6% of paediatric gastroenteritis and reaching the highest rate (30%) in 2006 (Figure 1). GI noroviruses were first detected in Palermo in 1994 and were found occasionally and scattered over the remaining study period; however, they were responsible for 3.2–7% of AGEs in 2002–2004 and 2011. The absence of noroviruses in the first years of surveillance in Sicily—from 1986 to 1988—could be ascribed to the low number of samples tested in 1986 and 1987; however, in 1988, a considerable number of faecal samples (182) tested negative for noroviruses. Although climate variations can affect the seasonal circulation of noroviruses, and long-term storage of samples can affect the stability of nucleic acids due to progressive degradation of viral RNA, our results could simply reflect the local viral epidemiology of that period, suggesting the introduction of noroviruses in Palermo only at the very end of the 1980s and their limited circulation until the mid-1990s. Alternatively, mutations in the primer/probe binding sites could have hindered the detection of the earliest norovirus strains using the molecular assays employed in this study. 

In this study, seasonal circulation was reliably calculated from 2012 onwards, when the number of samples collected was more abundant and homogeneous. Analysing the distribution of positive norovirus samples has shown a clear winter seasonality since 2011–2012, with peaks of norovirus circulation in November–January. Additional unexpected increases in positive norovirus samples were found in May of 2016 and 2019. A sudden reduction in the number of samples collected was observed from February 2020, with only occasional detection of noroviruses. Lower circulation of noroviruses was correlated with the COVID-19 pandemic and the consequent social distancing measures and use of personal protective equipment [43].

In order to investigate the genetic variability in GII noroviruses over time, Cap (ORF2) sequence analysis was performed, unveiling high genotype diversity until 1994, followed by the predominance of the GII.4 genotype from 1995 to 2020, with sporadic peaks of GII.3 and GII.2 genotypes in 2003–2004 and 2016, respectively (Figure 3a). The persisting epidemiological relevance of the GII.4 genotype in Palermo was characterized by a fast rate of evolution due to the accumulation of punctate mutations within the protruding (P) domain of the capsid (10^−3^ nt substitutions/site/year), coupled with intra- and inter-genotype recombination at the ORF1–ORF2 overlap in more recent years, starting with the Sydney strain in 2012. These mechanisms have been proposed for the effective selection of strains with improved fitness and the ability to evade the immune response [20,44]. As previously observed worldwide, nine pandemic variants of GII.4 noroviruses (Lordsdale, US95_96, Farmington_Hills_2002, Hunter_2004, Yerseke_2006a, Den Haag_2006b, Apeldoorn_2007, NewOrleans 2009 and Sydney_2012) emerged consecutively in Palermo [5,8,22,45,46], completely replacing each other every 2–3 years over the study period (Figure 4a). The first norovirus detected in this study (in 1989) was a GII.4 with a Cap gene genetically related to the Lordsdale genotype (99.8% nt identity) identified in the UK in 1993 (X86557) [47]. Lindesmith et al. hypothesized that pre-1995 Camberwell-like strains typically resulted in low-level endemic diseases in human populations, whereas since the mid-1990s, the accumulation of point mutations has promoted the spread of post-1996 Lordsdale/Grimsby strains [23]. However, the limited availability of norovirus sequences from the 1980s makes it difficult to date back the emergence of such an ancient genotype [48]. Recombination events were rarely detected in the older Italian strains, which usually carried their canonical GII.P4 polymerase, with the exception of a single GII.4_US95_96[GII.P2] strain detected in 1994. Conversely, in the last decade, sequential recombination events repeatedly affected the GII.4 Sydney_2012 variant. As already reported, this variant emerged in Italy in 2011 as a pre-epidemic strain containing the original GII.P31 polymerase, preceding the Australian and global circulation [19,40].

Thereafter, local circulation of the Sydney_2012 variant was sustained by the acquisition of a GII.P4 NewOrleans_2009 polymerase in 2013 and a GII.P16 polymerase in 2017 [35,49,50,51]. The sequential acquisition of such Pol genes may have been the key to the success of the Sydney variant and boosted its global emergence and spread.

The protruding P2 domain of the Cap protein possesses the epitopes involved in binding to the host cell and is responsible for virus antigenicity [21,52]. The P2 domain was sequenced to better understand the evolution of GII.4 norovirus strains over time. The aa alignment of 22 GII.4 Italian norovirus strains selected over the study period showed punctate mutations accumulating over time and were associated with the sequential emergence of GII.4 variants every 2–3 years. In particular, a conserved aa insertion at position 394 in Epitope D (amino acids 393–395), which is mostly a threonine residue, has been observed in all GII.4 strains since the emergence of the GII.4 Farmington_Hills_2002 variant in 2002 [53]. A change in residue 395 has been shown to alter the GII.4 norovirus antigenic profile [23]. Crystal structures of the putative Epitope D have shown its strategic position on the surface of the capsid, since this epitope interacts with the histo-blood group antigen (HBGA) binding site, suggesting the role of such mutations in both receptor switching and escaping herd immunity [54,55]. It was previously shown that the older GII.4 variants (i.e., Camberwell, Bristol, Lordsdale and US95_96) bound strongly only to antigen H of HBGA, while the new GII.4 variants extended their capability to also bind A and B antigens [55]. No aa changes were observed in epitopes A and E among the older Italian GII.4 strains with respect to the ancestral GII.4_Camberwell 1987 strain; however, since the detection of the GII.4_US95_96 variant, several aa changes have been observed in epitopes B, C and D. Interestingly, majority of the aa substitutions have accumulated since 2002, with emerging strain Farmington_Hills and mutation H395T representing key shifts in the antigenic milieu of GII.4 noroviruses. Since 2006, additional amino acid mutations have also been observed in Epitope A, located on the surface ridge of the capsid and probably involved in the evolution and adaptation of novel GII.4 variants [56]. The direct role of the escape phenotype of epitope A was further demonstrated by the Den Haag_2006b variant, which carries amino acid changes at positions 294, 296–298, 368 and 372 [9].

GII.3 noroviruses represented the second most relevant genotype detected in Palermo over the 35 years of surveillance, as also reported in other epidemiological studies [57,58]. Phylogenetic analysis of the Cap gene identified four different clusters (I–IV) of GII.3 strains [39]. Clusters I and II contained the oldest GII.3 strains, detected in the 1970s, 1980s and 1990s, while clusters III and IV included the strains circulating since the 2000s. In Palermo, GII.3 strains belonging to the four clusters described in the literature were detected over the study period, with clusters III and IV temporally overlapping from 2012 to 2016 and exclusive circulation of cluster IV thereafter. Italian GII.3 strains circulating from 1994 to 1997 in Palermo contained a P3 Pol gene. GII.3[P3] strains emerged globally in the 1980s and 1990s [59]. After 5 years of apparent absence from circulation (from 1998 to 2002), a succession of recombination events affecting GII.3 strains were detected in Palermo beginning in 2003, with the acquisition of P21, P12 and P16 Pol genes. The GII.3[P21] Cap/Pol combination represented one of the most successful GII.3 variants, being associated with symptomatic infections in children worldwide from 2000 to 2009 [60,61,62]. The increased mutation rate observed in the recombinant GII.3[P21] strains probably improved viral fitness [63]. As observed in Palermo, the progressive substitution of strains belonging to different Cap clusters and the acquisition of Pol genes due to recombination events possibly allowed the persistent detection of GII.3 strains. Recombinant GII.3[P21] strains were detected from 2003 to 2006 and then from 2013 to 2016 and in 2018, while GII.3[P12] strains circulated in 2012 and 2019 and GII.3[P16] strains circulated in 2011–2013. The latter strains were closely related to noroviruses detected in Parma (Italy) and Bangladesh in the same period [64,65]. 

GII.2 noroviruses represented the third most relevant genotype detected over the study period, with different Cap lineages and Cap/Pol combinations (Figure 4c). In particular, the strain GII.2[P2] circulated from 1990 to 1994 and again in 2011, whilst recombinant strains with polymerase GII.P34, GII.P4_2006b and GII.P16 appeared in 1996, 2009 and 2016, respectively. GII.2 noroviruses usually account for <1–1.5% of infections globally, with sporadic peaks in circulation [66,67,68,69,70]. Analysis of the Cap gene showed that the GII.2[P2] Italian strains collected in 2011–2016 were closely related to the GII.2[P16] Nashville strain (KY865307), which is supposed to be the donor of the polymerase for recombinant GII.4[P16] viruses [71,72]. Starting from 2011, the circulation of the GII.2 genotype in Palermo was sustained by a variety of strains containing two different Cap lineages and Cap/Pol combinations.

## 5. Conclusions

In conclusion, a unique 35-year collection of specimens was used to explore long-term trends in norovirus genetic diversity and evolution. Despite the large number of norovirus genotypes co-circulating in human populations, specific genotypes, GII.4 and GII.3, have predominated over time [9,39]. As already shown in previous studies, our findings confirm the predominant role of GII.4 Cap type starting from 1995; however, GII.3 and GII.2 retained a relevant epidemiological role over long periods, emerging and re-emerging over time with different Cap and Pol determinants. Noroviruses continuously and rapidly change, likely in order to escape the immunity elicited in a settled population, albeit in an intricate balance with host genetic resistance factors. GII.4 noroviruses also evolved to increase their binding affinities to HBGA receptors, with new epidemic strains exhibiting stronger binding intensities [27]. Although the GII.4 genotype is possibly the most successful norovirus strain due to its evolution ability, in this study, GII.3 and GII.2 noroviruses also persisted over time in a settled population through genetic evolution via accumulation of point mutations and Cap/Pol recombination. Studying the evolutionary dynamics of noroviruses can not only help to predict the emergence of new epidemic strains but is also pivotal to conceiving effective vaccines against noroviruses.

## Figures and Tables

**Figure 1 viruses-15-02303-f001:**
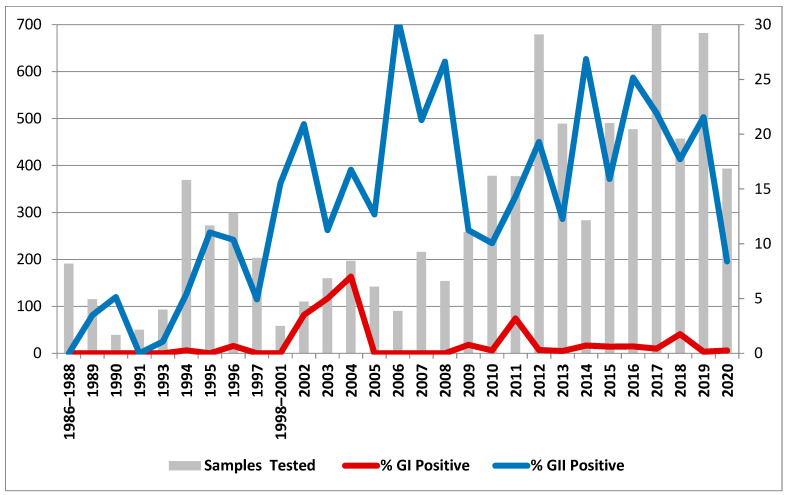
Prevalences of norovirus GI and GII infections in children hospitalized with AGE in Palermo, Italy, over the study period.

**Figure 2 viruses-15-02303-f002:**
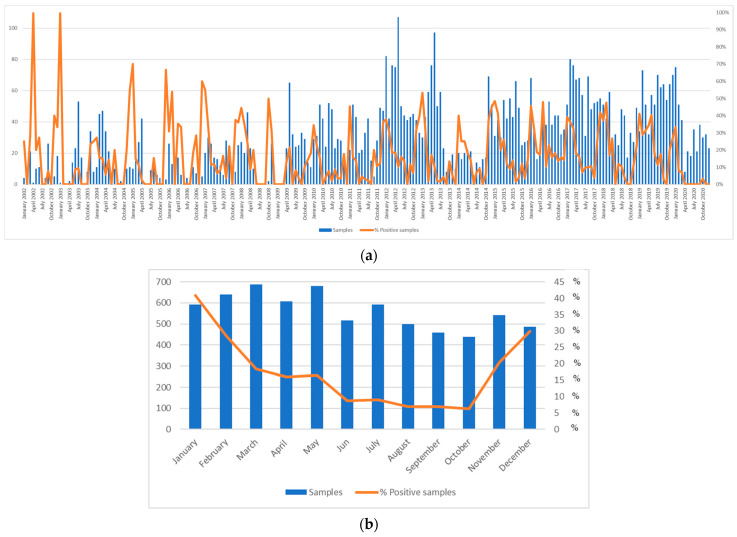
Monthly distributions and rates of norovirus positivity in samples collected from January 2002 to December 2020. Overall monthly collection of samples studied over the 19 years, from January 2002 to December 2020 (**a**) and cumulative monthly rates of norovirus positivity (**b**).

**Figure 3 viruses-15-02303-f003:**
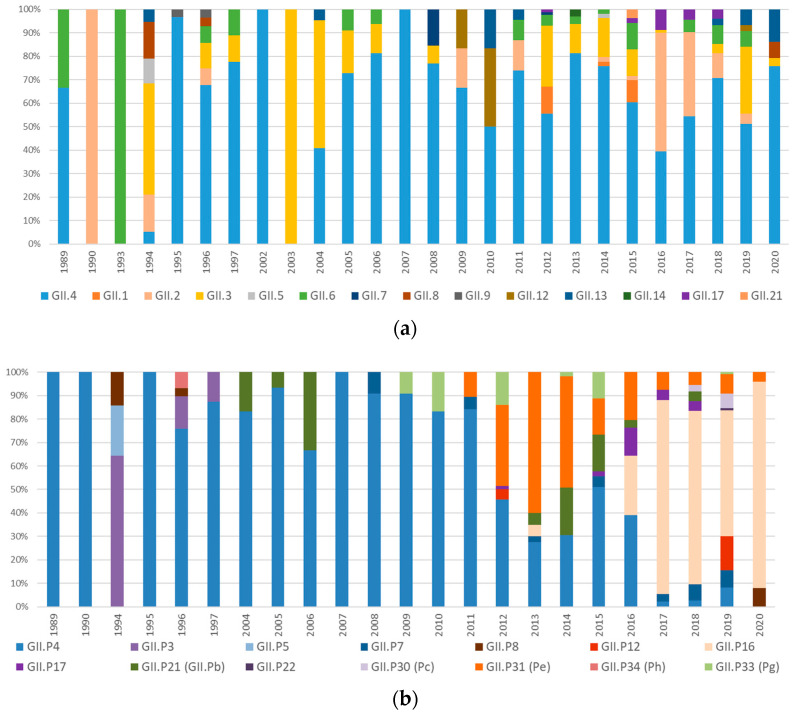
Temporal distribution of Cap (**a**) and Pol (**b**) norovirus genotypes and Cap/Pol genotype combinations (**c**) over the study period.

**Figure 4 viruses-15-02303-f004:**
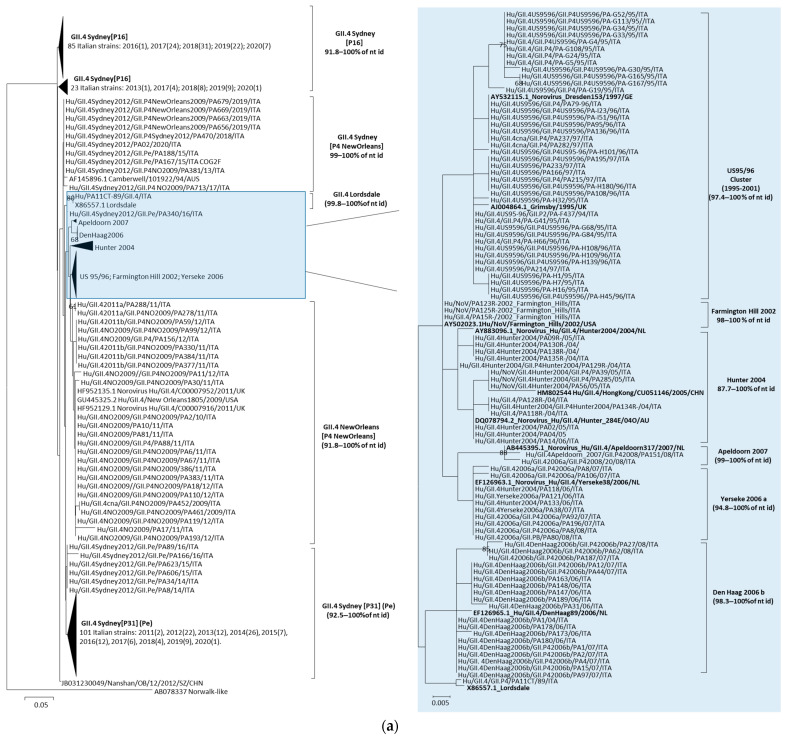
Phylogenetic analysis of partial ORF2 region in Italian GII.4 (**a**), GII.3 (**b**) and GII.2 (**c**) norovirus strains.

**Figure 5 viruses-15-02303-f005:**
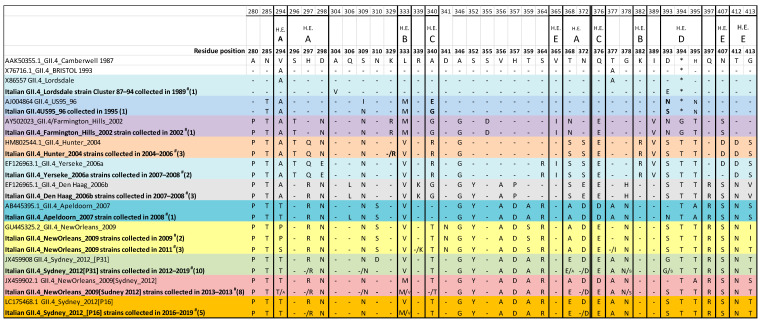
Evolutionary analysis of amino acids in representative GII.4 strains collected from 1987 to 2020. H.E, hypervariable Epitope; Amino acid substitutions in small characters were found in a minority of the sequences analysed. ^#^() Number of Italian GII.4 NoV strains analysed.

## Data Availability

The data are available from the corresponding authors under reasonable request.

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
