# Peer review of "Biological Specimen Banking as a Time Capsule to Explore the Temporal Dynamics of Norovirus Epidemiology"

_viruses, 2023, doi:10.3390/v15122303_

Round 1

Reviewer 1 Report

Comments and Suggestions for Authors

Bonura et al., tested archived samples between 1986 and 2020 from children hospitalized with AGE for norovirus and typed the positive samples. They found an overall detection rate of 17% (1.4% GI, 15.6% GII). Since very few sample collections from the 1980s and 1990s have been studied, this paper adds valuable info on the genotype trends over time and highlights that even 35 years ago, GII.4 viruses made up the majority of norovirus infections in children with AGE, which should be more articulated as conclusion of this work.

Throughout the manuscript, please don’t abbreviate norovirus to ‘NoV’. Single words in the English should not be abbreviated.

Abstract:

Line 20: replace “in the period” with ‘between”

Line 23: Please insert information on how many samples were tested by realtime RT-PCR and at what temperature they were stored.

Perhaps also report the overall prevalence of GII.4 as well as GII.2 and GII.3 in the abstract as that is part of what the authors report in the Discussion.

Introduction:

Page 2

Line 95: ‘epidemiological’ data are not generated. Since this is primarily a lab study what data are the authors referring to? Are they referring to the molecular epidemiology of norovirus across a 35-year time span perhaps?

Page 3

Line 116: Reference [19] refers to Eden et al., 2014 for the realtime RT-PCR assay that was used. This publication refers to ” A real-time RT-PCR targeting the 5′ end of NoV GI and GII capsid genes was employed using iQ SYBR Green Supermix (Bio-Rad, Hercules, US) and the primers COGIF/GISKR (Kageyama et al., 2003Kojima et al., 2002) and G2F3/G2SKR (Hansman et al., 2004Kojima et al., 2002) for NoV GI and GII detection, respectively” or to the assay that weas used by the New Zealand collaborators (Greening et al., 2012) mentioned in the Eden 2014 paper. Please clarify which assay was used and refer to the original publication first describing the assay.

Line 130/131. I typed the provided link but it seems it has been updated to https://calicivirustypingtool.cdc.gov 

Results:

Starting in 1995/1996, GII.4 viruses are classified as variant: See Kroneman et al., 2013 - Table 3. Please list the appropriate GII.4 variants throughout this manuscript.

Please report dual types (e.g., GII.4 Sydney[P31] etc) and do not report P-types separately as that is confusing. Lines 144-148 reports the % of each dual type which is how it should be reported.

The low prevalence rate of norovirus in the early years could also be explained by degradation of virus/viral RNA over time which the authors should acknowledge as a limitation.

Author Response

Reviewer #1

Bonura et al., tested archived samples between 1986 and 2020 from children hospitalized with AGE for norovirus and typed the positive samples. They found an overall detection rate of 17% (1.4% GI, 15.6% GII). Since very few sample collections from the 1980s and 1990s have been studied, this paper adds valuable info on the genotype trends over time and highlights that even 35 years ago, GII.4 viruses made up the majority of norovirus infections in children with AGE, which should be more articulated as conclusion of this work.

Reply: Conclusions have been revised providing more detailed explanation on the role of GII.4 viruses (page 15 lines 417-419).

R1.1    Throughout the manuscript, please don’t abbreviate norovirus to ‘NoV’. Single words in the English should not be abbreviated.

Reply to R1.1: The text has been modified as suggested.

Abstract:

R1.2    Line 20: replace “in the period” with ‘between”

Reply to R1.2: The text has been modified as suggested.

R1.3    Line 2: Please insert information on how many samples were tested by realtime RT-PCR and at what temperature they were stored.

Reply to R1.3: The information’s has been added to the “Abstract” section. The text has been modified as suggested.

R1.4    Perhaps also report the overall prevalence of GII.4 as well as GII.2 and GII.3 in the abstract as that is part of what the authors report in the Discussion.

Reply to R1.4: The information has been added to the “Abstract” section. The text has been modified as suggested.

Introduction:

R1.5    Page 2- Line 95: ‘epidemiological’ data are not generated. Since this is primarily a lab study what data are the authors referring to? Are they referring to the molecular epidemiology of norovirus across a 35-year time span perhaps?

Reply to R1.5: The text has been modified as suggested. 

R1.6    Page 3 - Line 116: Reference [19] refers to Eden et al., 2014 for the realtime RT-PCR assay that was used. This publication refers to ” A real-time RT-PCR targeting the 5′ end of NoV GI and GII capsid genes was employed using iQ SYBR Green Supermix (Bio-Rad, Hercules, US) and the primers COGIF/GISKR (Kageyama et al., 2003; Kojima et al., 2002) and G2F3/G2SKR (Hansman et al., 2004; Kojima et al., 2002) for NoV GI and GII detection, respectively” or to the assay that weas used by the New Zealand collaborators (Greening et al., 2012) mentioned in the Eden 2014 paper. Please clarify which assay was used and refer to the original publication first describing the assay.

Reply to R1.6: The reference 19 has been substituted with Kageyama et al., 2003, as suggested.

R1.7    Line 130/131. I typed the provided link but it seems it has been updated to https://calicivirustypingtool.cdc.gov

Reply to R1.7: The link has been replaced as suggested.

Results:

R1.8    Starting in 1995/1996, GII.4 viruses are classified as variant: See Kroneman et al., 2013 - Table 3. Please list the appropriate GII.4 variants throughout this manuscript.

Reply to R1.8: The text has been modified throughout the manuscript as suggested.

R1.9    Please report dual types (e.g., GII.4 Sydney[P31] etc) and do not report P-types separately as that is confusing. Lines 144-148 reports the % of each dual type which is how it should be reported.

Reply to R1.9: The % of each G/P combination of dual type was added and the confusing P-type report has been deleted as suggested.

R1.10  The low prevalence rate of norovirus in the early years could also be explained by degradation of virus/viral RNA over time which the authors should acknowledge as a limitation.

Reply to R1.10: The degradation of viral RNA has been acknowledge as a limitation in the discussion (page 13 line 303).

Reviewer 2 Report

Comments and Suggestions for Authors

This manuscript submitted to Viruses is a very interesting retrospective study on human norovirus molecular epidemiology in Palermo over the course of 35 years. The authors analyzed an extensive collection of stool samples from pediatric patients with acute gastroenteritis, detecting a variety of human norovirus strains, with 17% of all analyzed samples positive for norovirus.

This study is well conceived and executed. The remarkable amount of analyzed samples over such a long span of time gives a valuable insight into temporal norovirus evolution, which is undoubtedly of great relevance for the norovirus epidemiology field. In addition, the results are presented in a clear and detailed manner.

I particularly appreciate the fact that the authors didn’t limit the analysis to sequencing the conventional capsid and polymerase regions, but significantly enriched the depth and interest of their work by analyzing residue substitutions in the hypervariable P2 domain of the capsid protein among selected GII.4 variants (Fig. 4). This provides a real added value to this manuscript and could contribute to a future development of an efficacious norovirus vaccine.

My suggestion to make this work more complete and even more valuable to the norovirus research community - It would be very interesting to see if and how the period of sample collection correlates with the percentage of norovirus positivity. The authors could provide additional information (I assume it’s available - an additional table would be suitable) about when precisely (month, or at least season of the year) the samples were collected. For example, in a given year what percentage of samples collected during winter months was positive for norovirus compared to samples collected during summer/autumn/spring months? In your study, the highest percentage of HuNoV+ samples was detected in 2006, when less than 100 stool samples were collected - how does this compare to, for example, 2012, when almost 700 samples were collected, but the percentage of HuNoV+ samples was around 20%? You correctly acknowledge in the discussion that "climate variations may affect HuNoV seasonal circulation". Could you please address this observation?

Minor observations and questions:

1. Have you considered using an external virus extraction control to account for possible variable extraction efficiency and, above all, the presence of PCR inhibitors? (for example, adding a known concentration of Mengovirus).

2. Did you test pure (undiluted) RNAs only, or also diluted RNAs? Stool samples can present strong PCR inhibitors, therefore diluting the RNAs could sometimes help with genotyping, reducing the amount of false negative results. 

3. Please update the legend in Fig. 4 (add “residue position” and explain the meaning of “H.E.”).

Author Response

Reviewer #2

This manuscript submitted to Viruses is a very interesting retrospective study on human norovirus molecular epidemiology in Palermo over the course of 35 years. The authors analyzed an extensive collection of stool samples from pediatric patients with acute gastroenteritis, detecting a variety of human norovirus strains, with 17% of all analyzed samples positive for norovirus.

This study is well conceived and executed. The remarkable amount of analyzed samples over such a long span of time gives a valuable insight into temporal norovirus evolution, which is undoubtedly of great relevance for the norovirus epidemiology field. In addition, the results are presented in a clear and detailed manner.

I particularly appreciate the fact that the authors didn’t limit the analysis to sequencing the conventional capsid and polymerase regions, but significantly enriched the depth and interest of their work by analyzing residue substitutions in the hypervariable P2 domain of the capsid protein among selected GII.4 variants (Fig. 4). This provides a real added value to this manuscript and could contribute to a future development of an efficacious norovirus vaccine.

R2.1    My suggestion to make this work more complete and even more valuable to the norovirus research community - It would be very interesting to see if and how the period of sample collection correlates with the percentage of norovirus positivity. The authors could provide additional information (I assume it’s available - an additional table would be suitable) about when precisely (month, or at least season of the year) the samples were collected. For example, in a given year what percentage of samples collected during winter months was positive for norovirus compared to samples collected during summer/autumn/spring months? In your study, the highest percentage of HuNoV+ samples was detected in 2006, when less than 100 stool samples were collected - how does this compare to, for example, 2012, when almost 700 samples were collected, but the percentage of HuNoV+ samples was around 20%? You correctly acknowledge in the discussion that "climate variations may affect HuNoV seasonal circulation". Could you please address this observation?

Reply to R2.1: Additional information on seasonality has been added in figure 2 and in Results (page 3 lines 147-151). The limitation linked to differences in the abundance of stool sampling over the years has been acknowledge both in the text and in the figure legend.

Minor observations and questions:

R2.2    Have you considered using an external virus extraction control to account for possible variable extraction efficiency and, above all, the presence of PCR inhibitors? (for example, adding a known concentration of Mengovirus). Did you test pure (undiluted) RNAs only, or also diluted RNAs? Stool samples can present strong PCR inhibitors, therefore diluting the RNAs could sometimes help with genotyping, reducing the amount of false negative results.

Reply to R2.2: No external control was used but both pure (undiluted) and diluted (1:10) RNAs were used to reduce the effect of the possible presence of PCR inhibitors in stool samples. The latest information has been added in Material and Methods.

R2.3    Please update the legend in Fig. 4 (add “residue position” and explain the meaning of “H.E.”).

Reply to R2.3: The legend in the Figure 5 has been updated as suggested.